# Assessment of Complication Risk in the Treatment of Proximal Humerus Fractures: A Retrospective Analysis of 4019 Patients

**DOI:** 10.3390/jcm12051844

**Published:** 2023-02-25

**Authors:** Ralf Henkelmann, Pierre Hepp, Bastian Mester, Marcel Dudda, Philipp-Johannes Braun, Sebastian Kleen, Johannes Zellner, Michael Galler, Matthias Koenigshausen, Thomas A. Schildhauer, Tim Saier, Inga Trulson, Rony-Orijit Dey Hazra, Helmut Lill, Richard Glaab, Basil Bolt, Marcus Wagner, Michael J. Raschke, Jan Christoph Katthagen

**Affiliations:** 1Department of Orthopedics, Trauma and Plastic Surgery, Division of Arthroscopic and Special Joint Surgery/Sports Injuries, University of Leipzig, Liebigstrasse 20, 04103 Leipzig, Germany; 2Department of Trauma, Hand and Reconstructive Surgery, University Hospital Essen, Hufelandstraße 55, 45147 Essen, Germany; 3Department of Trauma and Orthopaedic Surger, BG Hospital Unfallkrankenhaus Berlin gGmbH, Warener Straße 7, 12683 Berlin, Germany; 4Sporthopaedicum Regensburg, 93053 Regensburg, Germany; 5Department of Trauma Surgery, Caritas Hospital St. Josef, Landshuter Strasse 65, 93053 Regensburg, Germany; 6Department of General and Trauma Surgery, University Bergmannsheil Bochum, Ruhr-University Bochum, Bürkle-de-la-Camp-Platz 1, 44789 Bochum, Germany; 7Department of Trauma Surgery, BG Trauma Center Murnau, 82418 Murnau, Germany; 8Institute for Laboratory Medicine, German Heart Center, Technical University of Munich, Lazarettstraße 36, 80636 Munich, Germany; 9Department for Orthopaedic Surgery and Traumatology, DIAKOVERE Friederikenstift, Humboldtstr. 5, 30169 Hannover, Germany; 10Department of Traumatology, Cantonal Hospital Aarau, 5001 Aarau, Switzerland; 11Institute for Medical Informatics, Statistics and Epidemiology (IMISE), Leipzig University, Härtelstraße 16-18, 04107 Leipzig, Germany; 12Department for Trauma, Hand and Reconstructive Surgery, University Hospital Muenster, Albert-Schweitzer-Campus 1, 48149 Muenster, Germany

**Keywords:** proximal humeral fracture, complication, operative treatment, nonoperative treatment, risk analysis

## Abstract

(1) Background: The treatment of proximal humeral fractures (PHFs) is debated controversially. Current clinical knowledge is mainly based on small single-center cohorts. The goal of this study was to evaluate the predictability of risk factors for complications after the treatment of a PHF in a large clinical cohort in a multicentric setting. (2) Methods: Clinical data of 4019 patients with PHFs were retrospectively collected from 9 participating hospitals. Risk factors for local complications of the affected shoulder were assessed using bi- and multivariate analyses. (3) Results: Fracture complexity with n = 3 or more fragments, cigarette smoking, age over 65 years, and female sex were identified as predictable individual risk factors for local complications after surgical therapy as well as the combination of female sex and smoking and the combination of age 65 years or older and ASA class 2 or higher. (4) Conclusion: Humeral head preserving reconstructive surgical therapy should critically be evaluated for patients with the risk factors abovementioned.

## 1. Introduction

To date, the treatment of proximal humeral fractures is debated controversially in the elderly population. Surgical treatment is challenging and clinical outcomes of randomized treatment studies indicate similar early outcomes after surgical and nonoperative treatment [1,2]. Humeral head-preserving reconstructive surgery is consistently associated with high complication rates despite the ongoing improvement of implants and surgical procedures [3,4]. Clinical studies with an evaluation of risk factors for complications and reasons for treatment failure are usually conducted on a retrospective basis in a single-center setting without evaluation of the predictability of the occurrence of complications [5,6]. Furthermore, studies evaluating the impact of fracture morphology on complication rates are often based on relatively small patient cohorts [7]. On the other hand, current studies evaluating risk factors from large-scale real-world health data lack clinical outcome information [6,8].

The goal of this study was, therefore, to re-evaluate risk factors for shoulder-related complications after treatment of proximal humeral fractures on a large clinical cohort in a multicentric setting and to evaluate their predictability.

## 2. Materials and Methods

### 2.1. Setting

We performed a retrospective study at 9 hospitals in Germany and Switzerland under the patronage of the trauma committee of the AGA (Society for Arthroscopy and Joint Surgery). A table with the individual patient numbers for the hospitals can be found in the Appendix A. The study protocol was approved by the leading Ethics Committee of the University of Leipzig (Reference number: 494/16-ek) and was confirmed by the corresponding ethics committees of all participating hospitals. The study was performed in accordance with the guidelines of the World Medical Association Declaration of Helsinki. The requirement for acquisition of informed consent from the patients was waived because of the retrospective nature of the study. All patients with proximal humerus fractures (PHF) aged 18 years and older, except patients with primary surgical treatment at another clinic or pathological fractures, were included.

All patients who were treated for PHF in the participating hospitals from January 2013 to December 2018 were identified by querying the hospitals’ databases using the International Classification of Disease code for PHFs (S42.20–S42.29). The medical records of all patients were screened manually to avoid the inclusion of patients who were improperly coded, were primarily operated on in another hospital, or did not meet our inclusion and exclusion criteria.

### 2.2. Data Description and Preparation

Primary data for 4038 fractures were available. For 13 patients with separately documented bilateral fractures and 1 patient with two separately documented consecutive fractures, one fracture was randomly selected. Further, we excluded 2 cases with ambiguously documented bilateral fractures and deleted 3 duplicated case documentations. This left us with 4019 patients, each with an unambiguously documented fracture. Data were carefully checked for consistency prior to statistical analysis. Contradictions were resolved by giving preference to more detailed information.

### 2.3. Patient Characteristics—Definitions of According to Covariates

Patient characteristics are shown in Table 1 and Table 2. Age was 18 years and older for 3989 patients and 8–17 years for 28 patients.

Comorbidities were defined as apoplexy with residuals in the history, arterial hypertension, asthma, congenital immune defects, chronic obstructive pulmonary disease (COPD), coronary heart disease, diabetes mellitus, emphysema, human immunodeficiency virus (HIV), kidney failure requiring dialysis, liver cirrhosis, pre-existing neurological conditions (e.g., multiple sclerosis), organ transplantation, rheumatic disease, or any type of tumor disease (counting a second tumor case as independent comorbidity). Substance abuse meant alcohol and/or drug abuse [9,10]. Alcohol or drug abuse was recorded as a parameter if it was diagnosed and reported at the time of treatment initiation of the proximal humerus fracture. Moreover, the ASA classification (ranging from 1 to 5) was recorded. Based on the social history, we defined three groups of patients: living alone/independent, partially dependent/attended, and fully dependent/residing in a nursing home.

In the risk analysis below, some of these covariates were binarized. Namely, we consider the binarized age (less than 65 ys. vs. 65 ys. and more), body mass index (BMI; less than 30 kg/m^2^ vs. 30 kg/m^2^ and more), ASA class (class 1 vs. all higher classes), comorbidities (less than 4 vs. 4 and more), and living situation (self-sustaining vs. supervised or nursing home).

### 2.4. Classification of Fractures and Treatments—Definitions of According Covariates

Fracture morphology was classified into the following groups by the attending surgeon based on the Codman classification and with the available X-ray and/or computed tomography images as follows: 2 part, 2 part greater tuberosity, 2 part lesser tuberosity, 3 part, 3 part greater tuberosity, 3 part lesser tuberosity, 4 part, head-split, and dislocation fractures (2 part, 3 part, 4 part, and head-split) [11]. Details are shown in Table 3.

Concomitant injuries were classified into none, not relevant (e.g., hematoma, abrasions), or relevant (e.g., other fractures, traumatic brain injury). In the risk analysis below, the concomitant injury of the affected extremity was binarized as none or not relevant vs. relevant, and the fracture pattern was binarized as 2-part (including 2-part greater tuberosity and 2-part lesser tuberosity) vs. all other patterns.

Concerning the treatment, we discerned two groups of patients. The first group comprised all patients who received operative therapy (N = 2557). The other group comprised all patients who received conservative therapy first regardless of whether a subsequent operation happened or not (N = 1462). Data about therapy are shown in Table 4.

### 2.5. Definition and Description of Complications

First, we documented the complications occurring in operative therapy. The following “complications of special type” were explicitly reported: postoperative complications, such as infection, screw perforation, implant dislocation, reduction loss/secondary dislocation, dislocation of the prosthesis, impingement, pseudarthrosis, and postoperative nerve damage as well as procedure-independent complications, namely, peri-implant fractures or new trauma after a fall, periprosthetic fracture, and humeral head necrosis. The complete data are shown in Table 5. All cases with reoperations are included in Rows 5 and 6.

In the following Table 6, complications after conservative therapy are summarized. In this case, any subsequent operation will be treated as a complication. In cases with no subsequent operation, we will consider only the “special complications” mentioned in Table 5 above.

### 2.6. Bivariate Analysis of Complication Risk

In a bivariate analysis of complication risk, we compared the following groups: (a) patients with explicitly reported absence of complications (Table 5, Row 1 or Table 6, Row 1) or mild complications in the sense of Table 5, Row 4, or Table 6, Row 4 (N = 2928) and (b) patients with explicitly reported complications and/or subsequent operation (Table 5, Rows 5 and 6, or Table 6, Rows 5 and 6) (N = 803). With respect to this binary dependent variable, we fitted bivariate logistic regression models, thus expressing the observed group differences in terms of odds ratios. All models included two independent variables: the initial treatment (conservative vs. operation) and a second covariate specified from the following list: sex, age (binarized), weight, BMI (binarized), ASA class (binarized), comorbidities (binarized), smoking, substance abuse, diabetes, immunosuppressive medication, living situation (binarized), injured side, trauma mechanism, concomitant injury of the affected extremity (binarized), or fracture pattern (binarized). Analysis was performed using R version 3.6.2. software (R Foundation for Statistical Computing: Vienna, Austria) [12].

### 2.7. Multivariate Analysis of Complication Risk

In-depth analysis of complication risk was performed by multivariate logistic regression. As binary dependent variables, we included the occurrence of complications as described in Section 2.5 above. Within all models, a single independent variable was fixed: the initial treatment (conservative vs. operation). The total number of patients with information about complications and initial treatment available was N = 3731. In order to prevent bias in following procedure of cross-validation, we included all 15 available covariates from Section 2.5 as possible further independent variables in the subsequent analysis.

For the generation and evaluation of the models, we pursued a strategy of *k*-cross-validation (*k* = 4) [13] (pp. 241 ff.). After excluding 3 patients with almost completely missing information, N = 3728 patients were included in the analysis (1240 of them with conservative and 2488 with operative treatment). By random subdivision of this sample, which was stratified for treatment, we generated 4 folds, A, B, C, and D, each consisting of N = 932 patients (thereof 310 with conservative and 622 with operative treatment in each case). From these folds, we built 4 combinations of training and validation sets (ABC/D, ABD/C; ACD/B, BCD/A) containing N = 2796 or N = 932 patients, respectively.

For every combination of training and validation sets, we completely enumerated all 2^15^ = 32768 possible models of the type
COMPLICATIONS ~ TREATMENT + COVARIATE_01 + …

Further, we considered the following 330 models with interaction terms:COMPLICATIONS ~ TREATMENT + COVARIATE_01 
+ TREATMENT: COVARIATE_01
COMPLICATIONS ~ TREATMENT + COVARIATE_01 + COVARIATE_02
+ TREATMENT: COVARIATE_01
COMPLICATIONS ~ TREATMENT + COVARIATE_01 + COVARIATE_02
+ TREATMENT: COVARIATE_02
COMPLICATIONS ~ TREATMENT + COVARIATE_01 + COVARIATE_02
+ COVARIATE_01: COVARIATE_02

All models were fitted by exclusively using the training data.

Model quality was assessed with the following quantities: (a) the significance of the chi-square omnibus test for the model, (b) the maximum of the significances obtained for the model’s coefficients, (c) the area under the curve (AUC) from receiver operating characteristic curve (ROC) analysis of the training set, (d) the correct classification rate (CC rate) for the training set (using sensitivity and specificity as given by the Youden index) [14], (e) the AUC for the ROC curve obtained by application of the model to the validation set, and (f) the CC rate obtained by application of the model to the validation set (where sensitivity and specificity are given by the Youden index from the training data). For every model, criteria (a) and (b) were maximized, and criteria (c)–(f) were averaged over all four combinations of training and validation sets. Analysis was performed using R version 3.6.2. software (R Foundation for Statistical Computing: Vienna, Austria) again [12].

For the subsequent optimization, we restricted ourselves to models with maximal global *p*-value less than or equal to 0.1 and maximal *p*-value for the coefficients less than or equal to 0.1. These obvious restrictions resulted in 11 models for further optimization. In particular, only a single model with interaction term possessed the required significance.

### 2.8. Optimization with Respect to AUC and CC Rate

Formally, the appropriate optimality definition for multicriterial optimization is so-called Pareto optimality [15]. This means that a model is considered Pareto optimal with respect to a number of criteria if one cannot find another model such that one criterion is improved but none of the remaining criteria is worsened. Obviously, any model which strictly maximizes a single criterion is Pareto optimal as well. In our case, Pareto optimization with respect to the four criteria (c)–(f), namely, mean AUC on training sets, mean CC rate on training sets, mean AUC on validation sets, and mean CC rate on validation sets, could be affected by simple comparison of tabulated values among all 11 feasible models.

In order to obtain predictions from a feasible model, we averaged the obtained coefficients and cutoffs over all four combinations of training and validation sets.

## 3. Results

### 3.1. Risk Factors for Complications: Bivariate Analysis

In a bivariate analysis of complication risk, we compared patients with and without complications, which we additionally correlated with treatment and a second covariate (for details, see Section 2.5). Results of bivariate risk analysis are shown in the following Table 7.

### 3.2. Risk Factors for Complications: Multivariate Analysis

The results of the multivariate risk analysis are shown in Table 8. Out of 32,768 models, only 11 models fulfilled the required global and coefficient significance threshold, see Section 2.6. For all feasible models, we document the maximal *p*-value for the global test and the maximal *p*-value for all coefficient tests over all four combinations of training and validation sets.

### 3.3. Optimal Models with Respect to AUC and CC Rate

In Table 9, we indicate the averaged values of criteria (c)–(f) over all four set combinations. In every column, the optimal value is printed in boldface.

Here, Model No. 1 maximizes simultaneously the mean CC rates (d) and (f), and Model No. 2 maximizes simultaneously the mean AUC values (c) and (e). Consequently, both Models Nos. 1 and 2 are Pareto optimal with respect to the four criteria (c)–(f) among the whole set of feasible models.

### 3.4. Predictors for Risk of Complications

#### 3.4.1. Averaged Model Data

The averaged coefficients and cutoffs for the feasible models are indicated in Table 10 below.

#### 3.4.2. How to Interpret These Models?

*(a)* For example, Model No. 5 makes the following prediction: If −1.645663 + 0.511489 * [Treatment: OP?]—0.375351 * [Age ≥ 65 y.?] + 0.284567 * [Fracture pattern: other than 2-part?] ≤ −1.224960, then complications are not to be expected. Otherwise, if −1.645663 + 0.511489 * [Treatment: OP?]—0.375351 * [Age ≥ 65 y.?] + 0.284567 * [Fracture pattern: other than 2-part?] > −1.224960, then complications are to be expected. The bracketed expressions, e.g., “[Treatment: OP?]”, take the value 1 if the answer is yes, but the value is 0 if the answer is no. Analogously, all other models except Nos. 3 and 11 are to be understood.*(b)* The only model involving a metric covariate is No. 3. Here, the prediction is: If −1.333256 − 0.339794 * [Treatment: OP?] + 0.078941 * Weight (in units of 10 kg) ≤ −1.061259, then complications are not to be expected. Otherwise, if −1.333256 − 0.339794 * [Treatment: OP?] + 0.078941 * Weight (in units of 10 kg) > −1.061259, then complications are to be expected.*(c)* Further, Model No. 11 with the interaction term is to read as follows: If −1.469708 + 0.794376 * [Treatment: OP?]—0.480485 * [Age ≥ 65 y.?] + 0.738945 * [ASA class: 2 or higher?]—0.855209 * [Treatment: OP?] * [ASA class: 2 or higher?] ≤ −1.272083, then complications are not to be expected. Otherwise, if −1.469708 + 0.794376 * [Treatment: OP?]—0.480485 * [Age ≥ 65 y.?] + 0.738945 * [ASA class: 2 or higher?]—0.855209 * [Treatment: OP?] * [ASA class: 2 or higher?] > −1.272083, then complications are to be expected.*(d)* For a model with two binary covariates, e.g., Model No. 1, the mentioned inequalities translate into a simple decision table. Indeed, if −1.850232 + 0.524765 * [Treatment: OP?] + 0.258821 * [Fracture pattern: other than 2-part?] ≤ −1.066647 implies the absence, and −1.850232 + 0.524765 * [Treatment: OP?] + 0.258821 * [Fracture pattern: other than 2-part?] > −1.066647 implies the presence of complications, then the four cases may be summarized in the following Table 11. An analogous interpretation is possible for Models Nos. 2, 4, and 8.

## 4. Discussion

The most important findings of this multicentric study were that the risk for local complications after surgical treatment of proximal humeral fractures is significantly and predictably increased independently, especially by (1) fracture complexity with n = 3 or more fragments, (2) smoking, (3) age over 65 years, and (4) female sex. Furthermore, the combination of (1) female sex and smoking and (2) age 65 years and older and ASA class 2 or higher were significantly predictive for local complications of the operated shoulder.

Since in this study, only 14% of patients with surgery for PHF were treated with shoulder arthroplasty, surgical treatment consisted of humeral head preserving reconstructive surgery in 86% of patients. The abovementioned most important findings can therefore be mainly related to humeral head-preserving surgical treatment options. When interpreting the results of this study, it must be kept in mind that regardless of the risk factors of fracture complexity, cigarette smoking, age over 65 years, and female sex, the quality of fracture reduction and retention is crucial for the avoidance of complications [16].

Most clinical outcome studies dealing with the treatment of proximal humeral fractures, regardless of whether retro- or prospective, have a single-center study design. The largest cohorts in the literature deal with about 1000 PHFs each [6,9]. When searching PubMed for [proximal humer* multicentric] in November 2022, 51 hits appeared. Since the PROFHER trial in 2015 with 250 patients treated either surgically or nonoperatively, only 4 further studies with multicentric design reported outcomes after humeral head preserving surgical treatment of PHFs [17,18,19,20,21]. These studies published in 2016 (two of them), 2017, and 2019 included between 56 and 127 patients. The cohort of our study with 4019 retrospectively analyzed patients is by far the largest one dealing with clinical outcomes of proximal humeral fracture treatment published in the literature up to date. In addition, this is the first study to evaluate the predictability of the named risk factors within a subset of the study cohort.

Although fracture morphology, smoking, age, and sex have been identified as risk factors for complications after surgical treatment of proximal humeral fractures before, this study is the first to underline these findings in a multicentric study and with additional evaluation of the predictive value of the named risk factors [7,8,22].

The fact that female sex was a risk factor for local complications of the operated shoulder is not completely congruent with the published literature. Koeppe et al. found that male sex was associated with higher mortality and increased risk for complications after surgical treatment of PHFs [23]. However, the local complication rate of the operated shoulder was only higher for men treated with reverse total shoulder arthroplasty but not for men treated with plate fixation.

Both treatment groups, as described in Section 2.3, differ substantially in their baseline parameters (this is not shown above). For this reason, univariate analysis of risk factors was inappropriate, and the treatment group had to always be included as a basic covariate. This holds for the bivariate as well as for the multivariate analysis. As the output of a *k*-cross validation procedure, the resulting multivariate models gain much more reliability than if they were based only on a single decomposition of the data set. The resulting models in multivariate analysis are comparatively simple but avoid overfit, thus allowing for better generalization to “unseen” data. This behavior is favored by the optimization criteria chosen here (AUC and CC rate). More complicated models have been excluded already by the prior bounds for significance. Although such models can be closer adapted to the given dataset, their predictive power in application to “unseen” data is inferior.

## 5. Limitations

The limitations of the study include the retrospective study design and the lack of a control group. The number of missing data was included in the tables. As is standard for bivariate or multivariate analyses, cases with missing variables were excluded.

## 6. Conclusions

Humeral head preserving reconstructive surgical therapy (open reduction and internal fixation) should critically be evaluated for patients with (1) high fracture complexity with n = 3 or more fragments, (2) cigarette smoking, (3) age over 65 years, and (4) female sex as individual risk factors or with the combination of (1) female sex and smoking and with the combination of (2) age 65 years or older and ASA class 2 or higher due to a high risk of postoperative complications of the operated shoulder. For patients with the risk factors abovementioned, nonoperative or reversed shoulder arthroplasty is strongly recommended.

## Figures and Tables

**Table 1 jcm-12-01844-t001:** Patient characteristics: part 1. m—meter, kg—kilogram, BMI—body mass index.

	Mean	Std. Dev.	Minimum	Maximum	Missing Values
Age (years)	66.9	17.3	8	101	2
Height (m)	1.68	0.10	1.28	2.05	1319
Weight (kg)	75.2	18.6	23.0	240.0	1291
BMI (kg/m^2^)	26.7	5.8	9.0	69.4	1322

**Table 2 jcm-12-01844-t002:** Patient characteristics: part 2. ASA—American Society of Anesthesiologists.

	Number	Percent
Sex: male/female	1267/2747	31.5/68.4
Sex: missing values	5	0.1
Age: 08–39/40–64 years	311/1326	7.7/33.0
Age: 65–79/80–101 years	1342/1038	33.4/25.8
Age: missing values	2	0.0
ASA class 1/2	499/1372	12.4/34.1
ASA class 3/4/5	1155/174/9	28.7/4.3/0.2
ASA class: missing values	810	20.2
Comorbidities: none/1 to 3	911/1966	22.7/48.9
Comorbidities: 4 or 5/6 and more	579/455	14.4/11.3
Comorbidities: missing values	108	2.7
Diabetes mellitus: yes/no	676/2878	16.8/71.6
Diabetes mellitus: unknown	465	11.6
Immunosuppressive medication: yes/no	102/3541	2.5/88.1
Immunosuppressive medication: unknown	376	9.4
Smoker: yes/no	483/2575	12.0/64.1
Smoker: unknown	961	23.9
Substance abuse: yes/no	320/2777	8.0/69.1
Substance abuse: unknown	922	22.9
Living situation: self-sustaining/supervision/nursing home	2817/276/267	70.1/6.9/6.6
Living situation: unknown	659	16.4

**Table 3 jcm-12-01844-t003:** Description and classification of fractures.

	Number	Percent
Injured side: right/left	1966/2035	48.9/50.6
Injured side: unknown	18	0.5
Trauma mechanism: low-energy/high-energy	3565/423	88.7/10.5
Trauma mechanism: unknown	31	0.8
Concomitant injury of affected extremity: none/relevant/not relevant	3137/371/317	78.1/9.2/7.9
Concomitant injury of affected extremity: unknown	194	4.8
Other concomitant injury: none/relevant/not relevant	2748/534/478	68.4/13.3/11.9
Other concomitant injury: unknown	259	6.4
Fracture pattern: 2-part	822	20.5
Fracture pattern: 2-part greater tuberosity	424	10.5
Fracture pattern: 2-part lesser tuberosity	44	1.1
Fracture pattern: 3-part	116	2.9
Fracture pattern: 3-part greater tuberosity	1348	33.5
Fracture pattern: 3-part lesser tuberosity	40	1.0
Fracture pattern: 4-part	630	15.7
Fracture pattern: head split	144	3.6
Fracture pattern: 2-part luxation	126	3.1
Fracture pattern: 3-part luxation	45	1.1
Fracture pattern: 4-part luxation	93	2.3
Fracture pattern: luxation with head split	42	1.0
Fracture pattern: unknown	145	3.6

**Table 4 jcm-12-01844-t004:** Description of the therapy. Data for reoperations are not included. RSA—reversed shoulder arthroplasty, HSA—hemi shoulder arthroplasty, n/a—not available.

	Number	Percent of Cases	Percent of Operations
Treatment: conservative	1462	36.4	n/a
Treatment: operation	2557	63.6	100.0
Treatment: missing values	0	0	n/a
Treatment strategy: plate	1670	41.6	65.3
Treatment strategy: intramedullary nail	392	9.8	15.3
Treatment strategy: HSA	51	1.3	2
Treatment strategy: RSA	299	7.4	11.7
Treatment strategy: screws	61	1.5	2.4
Treatment strategy: double plate	38	0.9	1.5
Treatment strategy: missing values	36	0.9	1.4

**Table 5 jcm-12-01844-t005:** Complications after operation (N = 2557). Note that for the special complications documented in Rows 7–17, multiple mentions of a single case are possible.

	Number	Percent of Cases	Percent of Complications
Complications after operation: no	1439	56.3	n/a
Complications after operation: yes	1051	41.1	100.0
Complications after operation: unknown	67	2.6	n/a
Complications: only mobility restrictions reported, no reoperation	440	17.2	41.9
Complications: other cases	218	8.5	20.7
Complications: at least one of the following problems reported (“special type”)	393	15.4	37.4
Postoperative infection	38	1.5	3.6
Screw perforation	102	4.0	9.7
Implant dislocation	81	3.2	7.7
Secondary dislocation	132	5.2	12.6
Luxation of prosthesis	19	0.7	1.8
Impingement	53	2.1	5.0
Pseudarthrosis	28	1.1	2.7
Postoperative nerve damage	21	0.8	2.0
Peri-implant fracture or new trauma	29	1.1	2.8
Periprosthetic fracture	11	0.4	1.0
Humeral head necrosis	93	3.6	8.8
Reoperation documented at all	527	20.6	50.1
Early reoperation within 12 months documented	353	13.8	33.6

**Table 6 jcm-12-01844-t006:** Complications after conservative therapy (N = 1462).

	Number	Percent of Cases	Percent of Complications
Complications: no	971	66.4	n/a
Complications: yes	270	18.5	100.0
Complications: unknown	221	15.1	n/a
Complications: neither of special type nor subsequent operation	78	5.4	28.9
Complications: special type but no subsequent operation	46	3.1	17.0
Complications: subsequent operation documented	146	10.0	54.1

**Table 7 jcm-12-01844-t007:** Bivariate analysis of risk factors, *p*-values, and odds ratios. Significant *p*-values (*p* ≤ 0.05) are printed in boldface.

Second Covariate	Treatment (Operation): *p*-Value	Treatment (Operation): Odds Ratio	Second Covariate: *p*-Value	Second Covariate: Odds Ratio
Sex: female	<0.0001	1.77	0.0019	0.77
Age, binarized: 65 ys. and more	<0.0001	1.76	<0.0001	0.70
Weight (numeric, 10 kg)	0.0048	0.71	0.0010	1.08
BMI, binarized: 30 kg/m^2^ and more	0.0033	0.70	0.0738	1.21
ASA class, binarized: class 2 and higher	0.2764	1.12	0.3835	0.90
Comorbidities, binarized: 4 and more	<0.0001	1.66	0.0290	1.22
Smoking: yes	<0.0001	1.60	0.0002	1.55
Substance abuse: yes	<0.0001	1.74	0.0119	1.42
Diabetes mellitus: yes	<0.0001	1.64	0.9901	1.00
Immunosuppressive therapy: yes	<0.0001	1.68	0.3811	1.23
Living situation, binarized: supervised or nursing home	<0.0001	1.93	0.7190	0.96
Injured side: right	<0.0001	1.79	0.0455	1.17
Trauma mechanism: high-energy	<0.0001	1.77	0.2653	1.15
Concom. injury of affected extremity, binarized: relevant	<0.0001	1.76	0.3730	1.13
Fracture pattern, binarized: other than 2-part	<0.0001	1.69	0.0056	1.30

**Table 8 jcm-12-01844-t008:** Multivariate risk analysis, all feasible models with *p* ≤ 0.1 in criteria (a) and (b) (N = 11). For the covariates, we use the following abbreviations: AGE—age (binarized), ASA—ASA class (binarized), COMORB—comorbidities (binarized), FRACT—fracture pattern (binarized), SEX—sex, SMOKER—smoking, TREAT—treatment, WEIGHT—weight.

Model No.	Covariates	Max. Global *p*-Value	Max. *p*-Value for Coefficients
1	TREAT, FRACT	<0.0001	0.0515
2	TREAT, SMOKER	<0.0001	0.0100
3	TREAT, WEIGHT	0.0064	0.0454
4	TREAT, AGE	<0.0001	0.0017
5	TREAT, AGE, FRACT	<0.0001	0.0317
6	TREAT, AGE, COMORB	<0.0001	0.0098
7	TREAT, AGE, COMORB, FRACT	<0.0001	0.0284
8	TREAT, SEX	<0.0001	0.0478
9	TREAT, SEX, FRACT	<0.0001	0.0416
10	TREAT, SEX, SMOKER	<0.0001	0.0639
11	TREAT, AGE, ASA, TREAT: ASA	0.0001	0.0317

**Table 9 jcm-12-01844-t009:** Rating of the feasible models by criteria (c)–(f). Covariates are denoted as in Table 8. AUC: area under the curve; CC: correct classification rate.

Model No.	Covariates	Mean AUC (Train.)	Mean CC Rate (Train.)	Mean AUC (Valid.)	Mean CC Rate (Valid.)
1	TREAT, FRACT	0.7507	78.51	0.7512	78.52
2	TREAT, SMOKER	0.7744	73.66	0.7743	73.65
3	TREAT, WEIGHT	0.5679	53.37	0.5690	53.38
4	TREAT, AGE	0.7339	67.55	0.7342	67.55
5	TREAT, AGE, FRACT	0.6866	67.59	0.6870	67.59
6	TREAT, AGE, COMORB	0.6821	59.43	0.6825	59.44
7	TREAT, AGE, COMORB, FRACT	0.6569	56.57	0.6567	56.57
8	TREAT, SEX	0.7387	70.03	0.7387	70.03
9	TREAT, SEX, FRACT	0.6848	70.15	0.6853	70.16
10	TREAT, SEX, SMOKER	0.7031	66.17	0.7030	66.16
11	TREAT, AGE, ASA, TREAT: ASA	0.7007	53.13	0.7004	53.12

**Table 10 jcm-12-01844-t010:** Averaged coefficients for the feasible models. In all models, the order of the covariates is the same as in Table 8 and Table 9.

Model No.	Intercept	Coeff_01	Coeff_02	Coeff_03	Coeff_04	Cutoff
1	−1.850232	0.524765	0.258821			−1.066647
2	−1.728730	0.471799	0.436402			−1.256932
3	−1.333256	−0.339794	0.078941			−1.061259
4	−1.492157	0.569173	−0.355059			−1.278044
5	−1.645663	0.511489	−0.375351	0.284567		−1.224960
6	−1.452566	0.494375	−0.440565	0.324396		−1.288617
7	−1.606725	0.428213	−0.462460	0.319434	0.297369	−1.343604
8	1.519835	0.571140	−0.263775			−1.212472
9	−1.673237	0.517943	−0.273121	0.273050		−1.428415
10	−1.515743	0.470628	−0.308434	0.370138		−1.353550
11	−1.469708	0.794376	−0.480485	0.738945	−0.855209	−1.272083

**Table 11 jcm-12-01844-t011:** Decision table arising from model no. 1.

	Treatment: No OP	Treatment: OP
Fracture pattern: 2-part	Complications: no	Complications: no
Fracture pattern: other than 2-part	Complications: no	Complications: yes

## Data Availability

Not applicable.

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
