# Peer review of "Assessment of Complication Risk in the Treatment of Proximal Humerus Fractures: A Retrospective Analysis of 4019 Patients"

_jcm, 2023, doi:10.3390/jcm12051844_

Round 1

Reviewer 1 Report

Authors assessed risk factors for local complications of the proximal humeral fractures using bi- and multivariate analyses. This seems to be relevant as "current studies evaluating risk factors from 67 large-scale real-world health data lack clinical outcomes information", as stated by authors. However, the study has several limitations which must be addressed.

There is a big confusion in the study. The TITLE of your article is "Assessment of complication risk in the treatment of proximal 2 humerus fractures: a retrospective analysis of 4019 patients" and the OBJECTIVE of your study is to re-evaluate risk factors for shoulder related complications after surgical treatment of proximal humeral fractures on a large clinical cohort in a multicentric setting and to evaluate their predictability. However, patients were divided into 2 groups, with 2557 treated surgically and 1462 treated initially non-surgically. Of these 1462 patients initially non operated, 146 were lately operated, as pointed out in Table 6 as "Complications: subsequent operation documented". Finally, authors present a multivariate analysis of complication risk based on 3728 patients (1240 non-operated and 2488 operated). Complications after non-operative treatment were unknown in 221 patients and omplications after operation were unknown in 67 patients. This is an extremely severe bias, generating a major confusion in the interpretation of your results.
I strongly suggest that you revise the research question, defining the participants of the study (patients treated surgically, patients treated non-surgically, or both; age; inclusion and exclusion criteria - why not exclude patients with unknown information?; and objective of your study. In addition, please revise all other aspects highlighted in my comments below.

# INTRODUCTION, Line 58: "Up to date, the treatment of proximal humeral fractures is debated controversially.". There is no debate or controversy between to operate on or not young patients sustaining a proximal humerus fracture. This controversy seems to be more active in the elderly population. Therefore, I strongly suggest authors to revise this sentence either establishing a population or re-writing it.

# MATERIAL AND METHODS, Line 79: Please, correct "Ethics Committee"; "s" is written in red.

# MATERIAL AND METHODS, Line 85: Please correct proximal humeurs fracture; it is written "humus".

# MATERIAL AND METHODS, Line 87: "... the International Classification of Disease code ...". It would be interesting to mention the code or codes used in the databases for identification of patients.

# MATERIAL AND METHODS, Lines 89-90: "... our inclusion and exclusion criteria.". Both inclusion and exclusion criteria were not mentioned before. It is important to mention these, please. In the next sentence authors mention that "All patients with PHF except patients with primary surgical treatment at another clinic or pathological fractures were included.", however what about age; did you include skeletally immature patients?

# MATERIAL AND METHODS, Line 95: "... was randomly selected.". How was the randomization done?

# MATERIAL AND METHODS, Line 98: "Contradictions were resolved ...". What did you consider to be a "contradiction"? Please, mention what was considered to be contradiction.

# MATERIAL AND METHODS, Lines 106-108: "Comorbidities were defined as apoplexy with residuals in the history, arterial hypertension, asthma, congenital immune defects, COPD, coronary heart disease, Diabetes mellitus, emphysema, HIV, kidney ...". Please write what means COPD, HIV, and BMI before using the abbreviation. This is the first time these terms are being mentioned in the text.

# MATERIAL AND METHODS, Line 111: "Substance abuse means alcohol and/...". What was considered alcohol abuse? 10 standard drinks per week, 0.08 grams of alcohol per week, or other? Please define. In addition, smoking was considered in your RESULTS section, however this is not mentioned here. Please, what was considered a smoker?

# TABLE 1: Table 1 is missing legends for the abbreviations, such as m; kg; kg/mˆ2; and Std.dev.

# TABLE 2: Table 2 is missing legends for the abbreviation for ASA.

# TABLES 4-6: Tables 4 to 6 needs to use a legend to explain what means n/a. Additionally, HSA is written in red in Table 4.

# CONCLUSIONS: "Humeral head preserving reconstructive surgical therapy should critically be evaluated for patients with (1) high fracture complexity with n=3 or more fragments, (2) cigarette smoking, (3) age over 65 years, and (4) female sex as individual risk factors or with the combination of (1) female sex and smoking and with the combination of (2) age 65 years or older and ASA class 2 or higher due to a high risk of postoperative complications of the operated shoulder.". Although authors mention that the major strength of their study is the large number of patients due to the multicentric study design, their conclusions do not differ substantially from previous single center studies, which showed that high-energy fracture patterns (4-part, head-split, and fracture-dislocations), tobacco use, older patients, and female patients are at higher risk of complications after surgical, but also non-surgical management. However, this is different from the conclusion presented in the ABSTRACT, where authors state that "Humeral head preserving reconstructive surgical therapy should critically be evaluated for patients with the risk factors above-mentioned". I suggest that after major revision in the title, objective, methodology, presentation of results, and discussion, authors can improve the conclusion. Even though it can be mentioned that "humeral head preserving reconstructive surgical therapy should critically be evaluated for patients with the risk factors above-mentioned", this is out of the scope of the study, therefore this has to be discussed properly in the DISCUSSION section, showing some alternatives to decrease the complication rate in these situations.

In my opinion, the study presents several limitations to be published, requiring MAJOR revision.

Reviewer 2 Report

A very good job, interesting and well prepared manuscript. I know, that only 14% of patients with surgery for PHF were treated with shoulder arthroplasty, and most important findings were mainly related to humeral head preserving surgical treatment options, BUT it would be interesting if tehrte is any difference in the risk of complications between these two surgeries. What were the indications to arthroplasty?
